Reliability of a standing isokinetic shoulder rotators strength test using a functional electromechanical dynamometer: effects of velocity

Martinez-Garcia Dario dariomg@correo.ugr.es 1
Rodriguez-Perea Angela 1
Barboza Paola 2
Ulloa-Díaz David 2
Jerez-Mayorga Daniel 3
Chirosa Ignacio 1
Chirosa Ríos Luis Javier 1
1 Department of Physical Education, Universidad de Granada , Granada , Andalucia , Spain
2 Department of Sports Sciences and Physical Conditioning, Catholic University of Most Holy Concepcion , Concepción , Bio Bio , Chile
3 Faculty of Rehabilitation Sciences, Universidad Andres Bello , Santiago , Santiago , Chile
Keogh Justin
Electronic publication date: 2020 Oct 27
Publication date: 2020
Volume: 8
Electronic Location ID: e9951
Received 2019 Oct 31; Accepted 2020 Aug 25
Copyright: ©2020 Martinez-Garcia et al.
Copyright year: 2020
Copyright holder: Martinez-Garcia et al.
License: This is an open access article distributed under the terms of the Creative Commons Attribution License, which permits unrestricted use, distribution, reproduction and adaptation in any medium and for any purpose provided that it is properly attributed. For attribution, the original author(s), title, publication source (PeerJ) and either DOI or URL of the article must be cited.
License URL: https://creativecommons.org/licenses/by/4.0/

Keywords: Isokinetic, Shoulder, Strength, Velocity

Funding: The authors received no funding for this work.

==============================
Background

The evaluation of the force in internal rotation (IR) and external rotation (ER) of the shoulder is commonly used to diagnose possible pathologies or disorders in the glenohumeral joint and to assess patient’s status and progression over time. Currently, there is new technology of multiple joint isokinetic dynamometry that allows to evaluate the strength in the human being. The main purpose of this study was to determine the absolute and relative reliability of concentric and eccentric internal and external shoulder rotators with a functional electromechanical dynamometer (FEMD).

Methods

Thirty-two male individuals (21.46 ±  2.1 years) were examined of concentric and eccentric strength of shoulder internal and external rotation with a FEMD at velocities of 0.3 m s−1 and 0.6 m s−1. Relative reliability was determined by intraclass correlation coefficients (ICC). Absolute reliability was quantified by standard error of measurement (SEM) and coefficient of variation (CV). Systematic differences across velocities testing circumstances, were analyzed with dependent t tests or repeated-measures analysis of variance in case of 2 or more than 2 conditions, respectively.

Results

Reliability was high to excellent for IR and ER on concentric and eccentric strength measurements, regardless of velocity used (ICC: 0.81–0.98, CV: 5.12–8.27% SEM: 4.06–15.04N). Concentric outcomes were more reliable than eccentric due to the possible familiarization of the population with the different stimuli.

Conclusion

All procedures examined showed high to excellent reliability for clinical use. However, a velocity of 0.60 m s−1 should be recommended for asymptomatic male patients because it demands less time for evaluation and patients find it more comfortable.

Introduction

Clinicians and researchers periodically assess strength changes in the status of their patients (McLaine et al., 2016). The evaluation of the strength in internal rotation (IR) and external rotation (ER) of the shoulder is commonly used to (I) diagnose possible pathologies or disorders in the glenohumeral joint, (II) evaluate the effectiveness and progression of the proposed treatment, and (III) as well as to quantify the change in muscle quality and the development of strength over time (Cools et al., 2014; Edouard et al., 2011). Also, objective measurements of this variable have proved to be essential for the identification of a patient at risk of injury, especially in athletes, against subjective measures or the self-report outcome score (Cools et al., 2015). Clinicians and researchers should have accurate and reliable protocol tests that objectively assess changes in strength over time that reflect the real gain or loss due to the protocol measurement (Edouard et al., 2011; Johansson et al., 2015).

The reliability measures in test-retest concern the repeatability of the data observed in a population in repeated measures (Hopkins, 2000). In medicine, it is a requirement to have relative reliability data with intraclass correlation coefficients (ICCs) and absolute reliability with a standard error of measurement (SEM) and coefficient of variation (CV) (Hopkins, Schabort & Hawley, 2001). The relative reliability indicates how well the rank order of the participants in the test are similar to the retest. The main problem with relative reliability is that it depends on the variability of the sample (Hopkins, 2000; Hopkins, Schabort & Hawley, 2001). However, absolute reliability is related to the consistency of individual scores (Hopkins, Schabort & Hawley, 2001; Weir, 2005).

Multiple tests have been created for the evaluation of shoulder strength with a wide variety of tools, including manual muscle testing (García, 2006; Lu et al., 2007), handheld dynamometers (Chamorro et al., 2017; Cools et al., 2014; Holt et al., 2016) or isokinetic devices (Andrade et al., 2010; Radaelli et al., 2010; Stickley et al., 2008). Isokinetic devices are considered as the gold standard in strength evaluations since they allow you to evaluate the maximum dynamic force throughout the range of motion (ROM) (Caruso, Brown & Tufano, 2012). Due to the biomechanics of the shoulder joint and its wide mobility, the reliability of the measurements in isokinetic devices are influenced by many factors (mechanical aspects, subjects, joints, and testing protocols), the assessment position, including the position of the shoulder (shoulder posture and joint-axis alignment) and the position of the body (sitting, supine, or standing and stabilization) (Broztman & Wilk, 2005; Edouard et al., 2011). Also, there are discrepancies in the literature regarding the most reliable velocities for testing. Additionally, most previous studies investigated the reproducibility of shoulder measurements between 60°/s and 240°/s (Nugent, Snodgrass & Callister, 2015). Higher velocities seem to be important when power assessment is necessary (the ability to produce moment rapidly) but less reliable. Some authors report faster velocities to be more reliable, while others report that low speeds are more reliable for measuring shoulder rotator’s strength (Edouard et al., 2011; Nugent, Snodgrass & Callister, 2015).

Nowadays, a new multiple joint isokinetic dynamometry has appeared (Dvir & Müller, 2019), denominated functional electromechanical dynamometry (FEMD). It allows us to evaluate the strength in the human being. The isometric strength of the shoulder rotators (Chamorro et al., 2018) and hip abductor strength (Cerda Vega et al., 2018) has been studied with this technology; however, shoulder maximal dynamic strength has not been studied yet (Chamorro et al., 2018) (Fig. 1).

Figure 1 FEMD dynasystem.

FEMD provides a quantified measurement of strength, and it is easy to use and has a lower cost compared to other isokinetic devices. Unlike others, this machine generates linear isokinetic speeds, dynamic modes (tonic, kinetic, elastic, inertial, conical) and statics modes (isometric, vibratory), allowing the evaluation and training through constant and/or variable resistance/velocity (Campos Jara et al., 2014). Furthermore, it has been described as a valid and reliable measurement method for evaluating upper and lower extremity muscle strength (Cerda Vega et al., 2018; Chamorro et al., 2018). This technology has been used to study the isometric strength of the shoulder rotators and has obtained high-reliability values (ICC > 0.94; CV < 10%) (Chamorro et al., 2018; Chamorro et al., 2019) In addition, this device has also been used to evaluate other joints such as lower limbs (Chamorro et al., 2017; Fàbrica et al., 2020) and trunk (Rodriguez-Perea et al., 2019).

Therefore, the main purpose of this study was (I) to determine the absolute and relative reliability of concentric and eccentric internal and external shoulder rotators in standing position with FEMD in the evaluation of the isokinetic strength to determine the most reliable test condition, and (II) To compare the absolute and relative reliability of different velocities for the assessment of isokinetic test. We hypothesized that (I) this new test will be a reliable method for the assessment of concentric and eccentric strength in shoulder rotators low speed would be more reliable than high speeds. The results are expected to provide new information regarding the dynamic shoulder strength evaluation protocols with FEMD.

Materials & Methods

Participants

Thirty-two asymptomatic male volunteers were recruited from local university (age: 21.46 ± 2.1 years, body mass: 69.22 ± 6.85 kg, height: 1.73.5 ± 0.07 m, Shoulder Pain and Disability Index (SPADI): 15.2 ± 3.8 and body mass index (BMI): 22.98 ± 1.607 kg/m2) without any experience in isokinetic or dynamometer devices participated in this study. Participants were eligible for the study if they were: (i) free of shoulder pain, with a maximum of 20% Shoulder Pain and Disability Index (SPADI); (ii) free of musculoskeletal injury; (iii) not practice specific training of upper body strength; and (iv) a maximum BMI of 25 kg/m2. All participants were informed regarding the nature, aims, and risks associated with the experimental procedure before they gave their written consent to participate. The study protocol was approved by the Biomedical Committee of the University (no 350/CEIH/2017) and was conducted in accordance with the Helsinki Declaration.

Study design

A repeated-measurement design was used to evaluate the shoulder rotator strength with different protocols. After two familiarization sessions, participants attended the laboratory on four separate days (at least 48 h apart) for two weeks. On each testing, day participants completed two sessions of familiarization and two protocol sessions with 0.3 m s−1 or 0.6 m s−1. Participants were asked to maintain their physical activity level during the two weeks of the study. All evaluations were conducted at the same time of the day (±1 h) for each participant and under similar environmental conditions (∼21 °C and ∼60% humidity). The order of velocities was randomly established.

Instrument

Isometric and isokinetic strength were evaluated with a FEMD (DynaSystem, Model Research, Granada, Spain) with a precision of three mm for displacement, 100 g for a sensed load, a sampling frequency of 1,000 Hz and a range of velocities between 0.05 m s−1 to 2.80 m s−1. Subjects can perform a wide variety of movements, and the device can deliver a wide variety of stimuli (isokinetic, isotonic, elastic, isometric, inertial, eccentric, and vibration). Its control core precisely regulates both force and angular velocity through a 2000 W electric motor. The user applies forces on a rope that winds on a roller, thus controlling and measuring both force and linear velocity. A load cell detects the tension applied to the rope, and the resulting signal goes to an analog-to-digital converter with 12-bit resolution. Displacement and speed data are collected with a 2,500 ppr encoder attached to the roller. Data from the different sensors are obtained at a frequency of 1 kHz (Fig. 2).

Figure 2 FEMD data screen.

Range of movement

Subjects were positioned standing and supporting the dominant upper limb on a subjection system of own manufacture (Fig. 3). The subjection system was regulated, taking into account the subject’s height with a variation of ±1 cm. The humerus was fixed with a cinch at 2/3 of the distance between the lateral epicondyle and the acromion. Both the position and the range of movement of the same was determined with a baseline goniometer (Gymna hoofdzetel, Bilzen, Belgium). The position consisted of a 90° adduction of the glenohumeral joint and a 90° flexion of the humero-ulnar joint. For the glenohumeral joint, the fulcrum was positioned in the acromion with the vertical arm stable and the arm movable along the humerus with the lateral epicondyle as a point of reference. For the humero-ulnar articulation, the fulcrum was positioned in the lateral epicondyle with the arm stable in horizontal and the arm movable along the forearm with the processus styloideus ulnae as a reference point. The range of movement was measured during the two familiarization sessions. The fulcrum is positioned in the olecranon with the stable arm in vertical (IR) or horizontal (ER) position and the mobile arm along the forearm with the processus styloideus ulnae as a reference until the goniometer is at an angle of 90° registering this distance to keep it stable throughout the different measurements.

Figure 3 Subjection system of own manufacture.

Familiarization protocol

Participants first attend (four subjects each time) to 90 min familiarization sessions with the FEMD and to testing the procedures. The familiarization consisted of a general warm-up for both test session consisted on 5 min of jogging (beats per minute < 130; measurement with a Polar M400), 5 min of joint mobility and 1 set of 5 repetitions of shoulder flexion-extension, five repetitions abduction and shoulder adduction, and five repetitions of IR and ER in the position of the test that was to be performed. After this, the same evaluation protocols that were going to be carried out in the evaluations were carried out.

Test protocol

Participants arrived in a well-rested condition at the start of each testing session. After the same warm-up of familiarization protocol, participants rested for 5 min before the initiation. The test consisted of two series of 5 maximum consecutive repetitions of shoulder rotators at a velocity of 0.60 m s−1 and 0.30 m s−1 and with the range of movement previously established. The rest between sets was a three-minute. The three highest repetitions of the mean force for the concentric contraction and the eccentric contraction were taken to calculate the mean dynamic force for each participant and velocity. The average strength of the total repetitions was taken into account in this measurement (Fig. 2).

Statistical analysis

The descriptive data are presented as mean ± SD. The distribution of the data was verified for the first time by the Shapiro–Wilk normality test. Reliability was assessed by t-tests of paired samples with the effect size ES, CV, and ICC, with 95% confidence intervals. The scale used for interpreting the magnitude of the ES was: negligible (<0.2), small (0.2–0.5), moderate (0.5–0.8), and large (≥0.8) (Cohen, 1988). Following Hopkins et al. (2009) classify through a qualitative scale the magnitude of the values of the ICC, being the values close to 0.1 low reliability, 0.3 moderate, 0.5 high, 0.7 very high, and those close to 0.9 extremely high. Sensitivity was estimated by the smallest detectable change (SDC) derived from the SEM (Courel-Ibáñez et al., 2019). The level of agreement between paired velocity outcomes from two measures was also assessed using Bland–Altman plots and the calculation of systematic bias and its 95% limits of agreement (LoA = bias ± 1.96 SD). Maximum errors (Max Error) at the 95% confidence interval were calculated from the Bland–Altman bias (Max Error) for the different condition outcomes analyzed (Paalanne et al., 2009). Reliability analyzes were performed using a customized spreadsheet (Hopkins et al., 2009) while the JASP software package (version 0.9.1.0, http://www.jasp-stats.org) was used for all other analyses. Figures were designed using GraphPad Prism 6.0 (GraphPad Software Inc., California, USA).

Results

Relative reliability for internal rotation test had high or excellent in all conditions (ICC: 0.81–0.93). Absolute reliability ranged from 6.31% to 8.27%; 5.58N to 6.8N in concentric contractions and 14.98N to 15.04N in eccentric contractions for CV and SEM respectively. ES was negligible in any of the conditions for internal rotation test (0.00–0.16) (Table 1).

Table 1 Reliability of the shoulder strength obtained during the isokinetic test on Dynasystem dynamometer.

Parameters	Mean ± SD	ES	ICC	95% CI lower-upper	CV (%)	SEM (%)	
			Session 1	Session 2						
Internal Rotation	0,3 m s−1	Concentric	83,10 ± 17.6	81,40 ± 16,1	−0,10	0,85	0,71–0,92	8,27	6,8	
Eccentric	206,50 ± 34,9	201,20 ± 31,8	−0,16	0,81	0,64–0,90	7,38	15,04	
0,6 m s−1	Concentric	88,80 ± 20,6	88,10 ± 19,6	−0,03	0,93	0,86–0,96	6,31	5,58	
Eccentric	218,00 ± 39,3	218,20 ± 41,6	0,00	0,87	0,75–0,93	6,87	14,98	
External Rotation	0,3 m s−1	Concentric	64,00 ± 13,2	67,00 ± 11,9	0,24	0,90	0,80–0,95	6,39	4,19	
Eccentric	198,90 ± 89	196,00 ± 89,9	−0,03	0,98	0,96–0,99	6,91	13,64	
0,6 m s−1	Concentric	64,40 ± 11,5	65,50 ± 12,1	0,10	0,89	0,78–0,94	6,26	4,06	
Eccentric	201,70 ± 32,6	194,90 ± 38,1	−0,19	0,92	0,85–0,96	5,12	10,16	
Notes.

SD Standard Deviation

ES Effect Size

ICC Intraclass Correlation Coefficient

CI Confidence Intervals

CV Coefficient of Variation

SEM Standard Error of Measurement

For external rotation tests, relative reliability test had high or excellent in all conditions (ICC: 0.89–0.98). Absolute reliability ranged from 5.12% to 6.91%; 4.06N to 4.19N in concentric contractions and 10.16N to 13.64N in eccentric contractions for CV and SEM respectively. ES was negligible in any of the conditions for external rotation test (−0.19–0.10), except for a small ES in concentric contraction in very slow velocity (ES: 0.24) (Table 1).

Bland-Altman plots showed the highest agreement and the most regular variation but exhibited more variation in eccentric contractions. Analysis of systematic biases by paired student t-tests found no significant difference except for external rotation in concentric contraction at 0.3 m s−1 and eccentric contraction at 0.6 m s−1 (Figs. 4 and 5).

Figure 4 Bland-Altman plots of test-retest for internal and internal rotation at 0,3 m s−1 and 0,6 m s−1.

Figure 5 Bland-Altman plots of test-retest for internal and external rotation at 0,3 m s−1 and 0,6 m s−1.

Discussion

The main purpose of this study was to determine the absolute and relative reliability of concentric and eccentric internal and external shoulder rotators in a standing position with a FEMD. To our knowledge, this was the first study to assess reliability for dynamic shoulder strength tests performed on a FEMD. The present investigation shows high to excellent reliability with ICC and CV values ranging from 0.81 to 0.93 and 6.31–8.27% for IR and from 0.89 to 0.98 and 5.12–6.91% for ER. Our findings confirm that this standing test is a reliable method for assessing shoulder rotation strength in asymptomatic adults. These results are of clinical relevance because shoulder rotation strength may be a useful marker of shoulder joint function.

Patients’ position in the evaluations of shoulder strength is a relevant variable which has to take into account for the reliability of the outcome of the test, due to the complex structure of the glenohumeral joint (Edouard et al., 2011; Forthomme et al., 2011). There is no scientific consensus about which the best position is to obtain reliable strength data in the shoulder test. The three positions most used by clinicians are supine (Andersen et al., 2017; Claudio Chamorro et al., 2018; Forthomme et al., 2011), sitting (Byram et al., 2010; Edouard et al., 2013; Radaelli et al., 2010) and standing (Frisiello et al., 1994; Greenfield et al., 1990). The sitting position has been the most widely studied due to the clinical nature of the studies and to an easier stabilization of the patient when evaluating (Edouard et al., 2011). Whether they are asymptomatic patients, athletes, or patients with chronic pain, the daily movements where the glenohumeral joint is used are usually not in this position (Beneka et al., 2002; Rodríguez-Rosell et al., 2017). Therefore, in this study, we propose the creation of a standing test, closer to the actual position of patients in the movements that encompass the shoulder. The results of this study show that standing tests can be performed reliably for asymptomatic patients. This study has obtained similar or better relative and absolute reliability data (ICC: 0.81 to 0.98, CV: 5.12 to 8.27) than most studies with this evaluation position Greenfield et al. (1990) (ICC: 0.81–0.95) and Frisiello et al. (1994) (ICC: 0.77–0.86) and similar data to articles with a sitting position for the same type of population: (Mayer et al., 1994) (ICC: 0.09 - 0.89; SEM: 2.8–9.2), (Kramer & Ng, 1996) (ICC: 0.83–0.96; SEM: 2–7) or (Malerba et al., 1993) (ICC: 0.44–0.90).

Second aim of the study was to compare the absolute and relative reliability of different velocities for the assessment of the isokinetic test. Velocity in isokinetic tests has been a widely studied variable in the literature (Andrade et al., 2010; Hadzic et al., 2012; Zanca et al., 2011). Although, most studies agree that speeds below 60°/s are usually the most reliable to perform strength assessments in most joints (Nugent, Snodgrass & Callister, 2015). There is no widely accepted consensus on which of these speeds is best suited for the shoulder joint (Edouard et al., 2011). In our study, we compared two speeds, one slow and one very slow to see if, as the velocity are lower, the reliability of the isokinetic test increased or, on the contrary, controlled slow speeds, but closer to the reality of the daily gestures of the patients produced a more reproducible application of force in these tests. Our results agree with this second idea, as previously noted in (Castro et al., 2017; Hadzic et al., 2012). While both velocities are reliable, the slow speed presents better results compared to the very slow one, as we can see in the Bland-Altman plots (Figs. 4 and 5), given that their bias are closer to 0 and most points are within the limits of agreement (LoA).

However, our analysis revealed large differences between the concentric and eccentric phases of both IR and ER. One possible explanation is that eccentric isokinetic tests require greater familiarization in IR while it was more reliable in the ER (Hadzic et al., 2012).

Bland–Altman bias (Max Error) for the different condition outcomes found no significant difference except for external rotation in concentric contraction at 0.3 m s−1 and eccentric contraction at 0.6 m s−1. This may be due to natural movements of the shoulder in the population tested, where the ER phase is predominantly eccentric, while the IR contraction is substantially concentric (Wagner et al., 2012). However, limits of agreement are wider in an eccentric phase like results obtained in (Pallarés et al., 2014), testing a stop between phases for a better adaptation of the subjects (Figs. 4 and 5). Clinicians and trainers should take these results into account to individualize the evaluation protocols according to the type of population and problem, whether they are overhead athletes or populations at risk of chronic pain in the shoulder.

Some limitations of this study should be noted to take them into account when evaluating our results. It was not possible to perform an inter-rater reliability analysis, so we can not know how this variable affects evaluations. We recommend both clinicians and coaches always have the same evaluator with the same patient. Also, the study was performed with asymptomatic active patients, which means that the results can not be extrapolated to other types of the population such as the sedentary population, or patients with chronic pain in the shoulder. Further study of these variables would be necessary to standardize the results to any type of population.

Conclusion

The study results show high to excellent reliability values for all procedures performed. Based on the results of this study, it may be reasonably concluded that the assessment in a standing position either at 0.6 m s−1 or 0.3 m s−1 are recommended because of practical applicability and body stabilization in asymptomatic patients. Clinicians are recommended to use more than one procedure to allow functional measurements based on the patient’s abilities at the moment of the evaluation.

Supplemental Information

Supplemental Information 1 Descriptive characteristics of the subjects

The raw data shows all data related to subjects, antropometric data and data of each trial of reliability test (without familiarization).

Click here for additional data file.

Supplemental Information 2 Realiability test trials

Click here for additional data file.

Additional Information and Declarations

Competing Interests

Author Contributions

Human Ethics

Data Availability

The authors declare there are no competing interests.

Dario Martinez-Garcia and Angela Rodriguez-Perea conceived and designed the experiments, performed the experiments, analyzed the data, prepared figures and/or tables, authored or reviewed drafts of the paper, and approved the final draft.

Paola Barboza conceived and designed the experiments, prepared figures and/or tables, authored or reviewed drafts of the paper, and approved the final draft.

David Ulloa-Díaz conceived and designed the experiments, performed the experiments, authored or reviewed drafts of the paper, and approved the final draft.

Daniel Jerez-Mayorga performed the experiments, analyzed the data, authored or reviewed drafts of the paper, and approved the final draft.

Ignacio Chirosa analyzed the data, authored or reviewed drafts of the paper, and approved the final draft.

Luis Javier Chirosa Ríos conceived and designed the experiments, analyzed the data, prepared figures and/or tables, authored or reviewed drafts of the paper, and approved the final draft.

The following information was supplied relating to ethical approvals (i.e., approving body and any reference numbers):

The study protocol was approved by the Biomedical Committee of the University of Granada (no 350/CEIH/2017).

The following information was supplied regarding data availability:

The raw measurements are available in the Supplemental Files.

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
