# Peer review of "Reliability of a standing isokinetic shoulder rotators strength test using a functional electromechanical dynamometer: effects of velocity"

_PeerJ, doi:10.7717/peerj.9951_

## Round 0.1 · original submission · Major Revisions

The two reviewers and I have a number of minor to major concerns with this manuscript, but have decided to give the author team an opportunity to respond to these concerns. Please make a concerted effort to take on board all these comments as this revision opportunity does not constitute likely acceptance of the paper, unless you can take on board these sources of feedback appropriately. One comment I have in particular is what actually is the severe lack of information regarding the FEMD. Can you please provide a picture of this device, its manufacturer details and greater detail of how this device is similar but different to the large commercial isokinetic dynamometry devices, including its relative potential advantages and disadvantages.

·

Basic reporting

Thank you for your appropriate selection of references. However, the following needs addressing for consistency:
An abbreviation must be written in full the first time it appears in the text. Line 56 requires this to be done for ICC’s, SEM or CV. Hence when described in lines 154 and 155 they can just have abbreviations. Line 67 FMD?
When using APA style of referencing (6th ed.) in text citations for three to five authors needs to be in full when first cited and then when reappears within text can be abbreviated to first author et al., date. This applies to lone 50, line 63, line 160, line 193, line 194, line 199, line 211.
The English language should be improved to ensure that an international audience can clearly understand your text. The current phrasing makes comprehension difficult from the section titled “familiarization protocol” through to the end of the paper. Other faults are detailed below:
Line 65 “Within these isokinetic devices are considered as the gold standard” does not make sense.
Line 80 “. . new technology that allows to evaluate the strength..”” is not correct English grammar.
Line 82 has numerals 4 and 3 inserted for no apparent reason.
Line 83 “ has been not studied yet” should read “ has not been studied yet”.
Line 89 “ . . for the assessment of isokinetic test.” An assessment is a test??
Line 91 delete “low speed would be more reliable….”as this is hypothesis (II).
Line 97 Should read “ Thirty-two asymptomatic male volunteers’. . . . “
Line 99 remove” s” from dynamometers.
Line 108 remove “s” from rotators.
Line 109 add “s” to session
Line 111 remove “s” from protocols
Line 119 add “d” to couple
Line 294 Strength is spelled incorrectly.

Experimental design

The introduction must include further background for the reader regarding the equipment used for this study, being the Functional Electromechanical Dynamometer. I suggest you expand on the description/background of your device. Currently, there is only one sentence reporting this in Line 80 “new technology that allows evaluation of the human being in the Functional Electromechanical Dynamometer (Dvir & Muller, 2019), with only isometric strength of the shoulder rotators and hip abductor strength previously studied."

As the reviewer of this manuscript I then read the reference provided by Dvir and Muller, 2019 and established the following:

"These dynamometers typically consist of an electrical motor coupled to a linear transmission, which moves a slider with footplates or special handles. For dynamic action, MIDs (Multiple joint Isokinetic Devices) enable either concentric or eccentric exertions, and the basic input parameters are therefore the range of motion (in millimeters or centimeters) and the velocity (in m per sec or cm per sec), whereas the main output parameter is the peak force (PF) in Newton (N). Multiple-joint isokinetic dynamometers are used especially for testing and conditioning of coupled ankle-knee-hip extension, known as leg press (LP), coupled shoulder flexion and elbow extension (or their reciprocals), and its reverse action, known as bench press (BP) and vertical lifting.
It appears the device used in this study is: referred to as the unconstrained design, provides concentric or eccentric resistance to a free, multiarticular whole-body motion, using a cable or a rod without providing proximal stabilization. The stand-alone cable variant, manufactured by Haefni Health Systems (Granada, Spain), incorporates a spool around which a cable is wound. In concentric exertions, the cable is normally drawn away from the spool, whereas in eccentric actions, the opposite cable spool motion is taking place."

Validity of the findings

Thank you for the appropriate use of test protocol and statistical analysis. However, further detail is needed in the methods section of your paper to make clear to the reader the validity of your findings.
As the reviewer of this manuscript I read the reference provided by Dvir and Muller, 2019 and established the following:

"By virtue of their design and operation, MIDs offer a more functional approach. Nevertheless, users of MIDs should be aware of the implications associated with scaling, specifically that a constant linear velocity will not result in a concomitant constant angular joint velocity and therefore a simple comparison is not valid. Furthermore, for the same linear velocity, people of taller stature will experience lower angular joint velocity compared with their shorter counterparts and thus, as far as concentric efforts are concerned, strength capacity might be overestimated in taller athletes or patients. However, this discrepancy is especially pertinent to the issue of reference values (norms), where strict uniform testing and conditioning must apply to all participants. Given the generally high reproducibility of MID-derived findings as indicated by well-conducted studies, ensuring same input parameters such as RoM, velocity, and participant positioning should be quite sufficient for valid data interpretation in terms of bilateral differences or changes in the course of rehabilitation or training. By contrast, inter-individual comparisons will require consideration of the previously described limitations."

Therefore, confirmation of the following is required to confirm the validity of the findings:

Why is “the pulley system and subjection system of own manufacture” – detailed under Instrument. This greatly affects the validity of these findings. Are they the same as those used in the previous studies quoted looked at isometric shoulder testing and hip abduction testing?

Why were the subjects positioned standing or sitting to determine range of movement when all testing was performed in standing. This raises the concern that variances in height of participants has skewed the results.

A 90 minute period of familiarization seems excessive. Although the participants recruited had no experience in the use of dynamometry, this time allows each participant to have learned and practised the required tests to an “expert” level.

Dvir and Muller (2019) report this "functional electromechanical dynamometry is for testing coupled movements such as shoulder flexion and elbow extension (otherwise known as bench press)". Why then was only shoulder rotation reported?

Interestingly, when citing previous isokinetic dynamometry studies which used the standing position they were thirty years ago whereas studies cited for supine and sitting positions are within the past ten years. This raises the question of why standing has not been continued as a testing position.

Additional comments

Thank you for your well constructed reliability study, with appropriate test protocol and statistical analysis, even though it has very limited external validity. Before considering this manuscript as suitable to be recommended for publication , you need to substantiate the validity of your study by further explanation on the equipment used and explanation on methods.

Reviewer 2 ·

Basic reporting

The language could be improved.
The introduction demonstrates how the work is related to the broader field of knowledge.
The structure fits the Journals' instructions.
Although the figures are relevant to the content, some important information (which could be presented as figures) are missing.
Row data have been made available.

Experimental design

The presented research fits the Journals' scope.
The research question is limited to the reliability of the novel device, without previous information about the concurrent validity. In addition, test-retest has been explored without providing information about within-session reliability.
Some parts of the methods sections should be presented in more detail (see general comments)

Validity of the findings

The impact and novelty of the study are limited (see general comments).
The conclusions are questionable (see general comments).
In some part of the discussion, speculations are made without evidence based on the presented results.

Additional comments

GENERAL COMMENTS
Thank you for the opportunity to review the manuscript entitled “Reliability of a standing isokinetic shoulder rotators strength test using a functional electromechanical dynamometer: Effects of velocity”. The authors aimed to determine the absolute and relative reliability of concentric and eccentric internal and external shoulder rotators with a functional electromechanical dynamometer (FEMD). They concluded that all procedures examined showed high to excellent reliability for clinical use. They recommended that the velocity of 0.60 m·s-1 should be used for asymptomatic male patients because it demands less time for evaluation and patients find it more comfortable.
The topic addressed in the manuscript is relevant to the journal’s readership. Although the structure of the manuscript follows the Journal instructions, the main limitation of the article is related to the fact that the reliability of the novel device has been explored without prior reports regarding its validity. Also, within-session reliability was not explored.
SPECIFIC COMMENTS
Abstract
Line 22: In: “… relative reliability of concentric and eccentric internal and external shoulder rotators …” should be changed to: … relative reliability of concentric and eccentric internal and external strength of shoulder rotators …”
Introduction
Line 43: The first sentence seems to be incomplete: What is periodically assessed?
Line:
Methods
Since the main aim of the article was to explore reliability of the novel device, the figure of the test setup, as well as the signal profiles (force, velocity and angle over time) through the range of motion and at various constant velocities, are necessary to illustrate the factors (biomechanical conditions) which could affect the strength measures and therefore the explored reliability.
Line149-150: “The three highest repetitions of the mean force for the concentric contraction and for the eccentric contraction were taken to calculate the dynamic force.” Please explain how the dynamic force was calculated (average?).
Results
Please elaborate results in more details.
Discussion
The findings regarding systematic bias are not discussed.
Line 188-189: The position could not be reliable, only the test outcome obtained in a certain position.
Line 197-202: This is an interesting point. However, due to a large number of degrees of freedom, without fixation, it’ hard to isolate muscles responsible for the specific movement. Please comment.
Line 202-207: See the general comment regarding the validity.
Line 209: At the beginning of the sentence please substitute “Other” with “Second”.
Line 209-222: One of the main advantages of the isokinetic dynamometry in general, is the ability to assess partly independent mechanical qualities of the tested muscle through the range of different constant velocities (Van Roie et al. 2013, Grbic et al 2017). Therefore suggesting the application of only one condition due to higher reliability (although the authors are claiming that other application of other velocities are highly reliable) is questionable.
REFERENCES:
Van Roie, E., Bautmans, I., Boonen, S., Coudyzer, W., Kennis, E., & Delecluse, C. (2013). Impact of external resistance and maximal effort on force-velocity characteristics of the knee extensors during strengthening exercise: a randomized controlled experiment. The Journal of Strength & Conditioning Research, 27(4), 1118-1127.
Grbic, V., Djuric, S., Knezevic, O. M., Mirkov, D. M., Nedeljkovic, A., & Jaric, S. (2017). A novel two-velocity method for elaborate isokinetic testing of knee extensors. International journal of sports medicine, 38(10), 741-746.

---

## Round 0.2 · accepted · Accept

I appreciate the hard work in attending to the reviewers comments and am happy to recommend the paper be accepted for publication.

·

Basic reporting

No comment

Experimental design

No comment

Validity of the findings

No comment

Additional comments

Thank you for taking the time to edit your original submission. The English language is much improved. Detail on the equipment used and methodology is now clear and succinct.

Reviewer 2 ·

Basic reporting

The authors have improved the manuscript substantially.

Experimental design

All details regarding the experimental design are sufficiently addressed.

Validity of the findings

The findings are well stated and linked to the results and the research question.

Additional comments

No further comments.